# The Effect of Exercise on Reducing Lymphedema Severity in Breast Cancer Survivors

**DOI:** 10.3390/cancers16071367

**Published:** 2024-03-30

**Authors:** Yanxue Lian, Simran Sandhu, Yamikani Asefa, Ananya Gupta

**Affiliations:** Department of Physiology, University of Galway, H91-TK33 Galway, Ireland; y.lian5@nuigalway.ie (Y.L.); s.sandhu1@nuigalway.ie (S.S.); y.asefa1@nuigalway.ie (Y.A.)

**Keywords:** breast cancer-related lymphedema, exercise, physical activity, yoga, resistance exercise

## Abstract

**Simple Summary:**

Some cancer survivors, especially breast cancer survivors, may face the issue of lymphedema. Although it tends to be incurable, there are some methods to manage lymphedema. Exercise was thought to aggravate lymphedema severity in cancer survivors until a completely opposite view was proposed in 1998. Since then, an increasing number of studies have been carried out to investigate the effects of exercise on cancer survivors with lymphedema. Currently, exercise is widely considered safe and beneficial. However, the formulation of exercise prescriptions for patients with lymphedema has been under exploration. The aim of this literature review is to extract information from existing studies to provide some advice for the design of exercise prescriptions in terms of reducing lymphedema severity. The findings of this literature may help patients reduce the severity of lymphedema and improve their quality of life. It may also help reduce healthcare costs and improve affordability.

**Abstract:**

Exercise has been repeatedly shown to be safe and beneficial for cancer survivors. However, there is no normative guideline for exercise prescription, and it is still under exploration. Therefore, this literature review aims to provide some advice for the formulation of exercise prescriptions for patients with breast cancer-related lymphedema (BCRL) from the perspective of reducing lymphedema severity. A review of relevant studies published before November 2023 was conducted using three scientific databases: PubMed, Embase, and Scopus. A total of 2696 articles were found. Eventually, 13 studies fulfilled the inclusion criteria and were included in this literature review. We concluded that daily, or nearly daily, exercise at home can be recommended. Moreover, reduced lymphedema severity may not be maintained after ceasing the exercise program, so exercise should be a lifelong practice.

## 1. Introduction

Breast cancer is the most common cancer in females and the most common cancer overall [1]. Recent statistics reported by the World Health Organization indicate that 2.3 million women were diagnosed with breast cancer, and 685,000 of them passed away globally in 2020, with 7.8 million women diagnosed with breast cancer in the past 5 years [2]. The five-year relative survival rate for women with localized breast cancer can be up to 99% with the improvement of early detection, according to statistics from diagnosis between 2011 and 2017 [3]. However, they may face a challenge: lymphedema. Some researchers believe that 20% of breast cancer survivors suffer from lymphedema [4,5]. In fact, the data is unclear, and the percentage could be much higher. One study showed that lymphedema occurs in 94% of breast cancer survivors based on the various methods, timing, and criteria of diagnosis, as well as the design of studies [6]. Moreover, lymphedema can emerge at any time after the initial treatments, and this risk continues to the end of the patient’s life [7]. Due to the high incidence of breast cancer and its related lymphedema, more exploration is critical.

Lymphedema is defined as the inability of the lymphatic system to transport lymph fluid out of the affected area, resulting in an abnormal accumulation of protein-rich fluid in the interstitial tissues [8,9]. Generally, it can be divided into two types: primary and secondary lymphedema [10]. BCRL is the most prevalent form of secondary lymphedema [10]. During cancer treatment, the lymphatic system can be partially removed, disturbing the filtration and normal flow of the lymphatic fluid so that it accumulates in the lymphatic vessel and finally exits to interstitial tissues through the vessel wall [11].

It is important to note that lymphedema is incurable and chronic [12]. Fortunately, there are some effective management methods available, and they are more likely to be useful at the early stage than late stage [13]. Although some recent studies show that surgical treatments may be superior to non-surgical treatments to some extent, non-surgical treatments are still emphasized as indispensable [14,15]. Moreover, surgical treatments may be suggested when non-surgical treatments cannot control the symptoms, especially in the early stage of lymphedema [13]. Non-surgical treatments consist of complete decongestive therapy (CDT), manual lymph drainage (MLD), compression therapy, compression garments, skin care, advanced pneumatic compression pumps, exercise, and laser therapy [14]. Of those, CDT, also called complex decongestive physiotherapy (CDP), which includes MLD, bandaging, compression, skin care, and exercise, has been most widely used as a standard treatment [10,16,17]. However, exercise, as part of the standard management, is far less understood because it was thought to be unsafe for breast cancer survivors to be involved in upper body exercise, as it could induce and aggravate lymphedema [18].

In 1998, a study of dragon boat racing as a milestone was reported to suggest that exercise might play an important role in recovering from BCRL [19]. Since then, a series of studies with exercise interventions in cancer survivors with lymphedema have been conducted, followed by literature reviews. The present literature reviews have predominantly focused on determining the effects of a certain type of exercise, such as resistance training [12,20], yoga [21,22], and water-based exercise [23], on cancer survivors with lymphedema, and exercise has been repeatedly shown to be safe and beneficial for cancer survivors. However, there is no normative guideline for exercise prescription, and it is still under exploration. Therefore, this literature review aims to provide some advice for the formulation of exercise prescriptions for patients with BCRL from the perspective of reducing lymphedema severity.

## 2. Methods

A review of relevant studies published before November 2023 was conducted using the most up-to-date versions of three scientific databases PubMed (2.0) (https://pubmed.ncbi.nlm.nih.gov/, accessed on 11 November 2023), Embase (https://www.embase.com/, accessed on 12 November 2023), and Scopus (https://www.scopus.com/, accessed on 12 November 2023). The key terms searched were “breast cancer” AND “lymphedema” OR “lymphoedema” AND “exercise”. Further searching terms were “weightlifting” OR “physical activity” OR “Yoga” OR “Pilates” OR “resistance training” OR “swimming” OR “strength training” OR “aerobic exercise” OR “water-based exercise” OR “aqua therapy” and possible variations of each exercise type. The inclusion criteria were: (i) randomized controlled trials (RCTs), (ii) inclusion of adult participants (>18) living with BCRL, (iii) peer-reviewed and published in the English language, (iv) completed studies with objective measurement of lymphedema severity as one of the outcomes reported, and (v) an organized exercise plan. The exclusion criteria were: (i) some or all participants at high risk of lymphedema, (ii) randomized cross-over control trials, (iii) articles without English translations, (iv) studies with experimental study variables (such as weight control through dietary management) other than exercise but the effects of exercise cannot be isolated, (v) ongoing experimental studies, (vi) physical therapy as the intervention, (vii) the control group was given any kind of exercise interventions except for self-care or standard care.

A total of 2696 articles were found. After the removal of duplicates, 1587 articles were eligible. The process of selection could be divided into three stages, carried out by two researchers independently. Most of these articles could be excluded by titles and abstracts, and 101 articles entered the second stage—the full-text review. Controversial results were discussed for inclusion. Eventually, 15 RCTs were identified, but lymphedema severity was not measured objectively in 2 studies, so finally, 13 studies fulfilled the inclusion criteria and were included in this literature review (Figure 1).

## 3. Results

The risk of bias in these studies was analyzed by Review Manager 5.4.1. (Figure 2). It showed good methodological quality with a low or unclear risk of bias in 11 articles overall, but 2 studies had a high risk of bias (Figure 2). Moreover, the modalities of exercise were different among these studies: participants in two studies performed yoga, participants in five studies performed resistance exercise alone, participants in four studies performed a combination of aerobic and resistance exercise, and participants in two studies performed water-based exercise. The duration of the interventions ranged from 8 weeks to 1 year. The frequency of exercise fluctuated between once per week and daily. The number of subjects ranged from 14 to 177. The outcome that is of primary concern in this literature review, lymphedema severity, was reflected by different indicators and measured by different methods: limb volume by water displacement or Perometer, limb circumference by non-stretch tape, extracellular fluid by Bioimpedance spectroscopy (BIS), and tissue composition by dual-energy X-ray absorptiometry.

### 3.1. Yoga

Two RCTs with yoga intervention by Loudon et al. (2014) and Pasyar et al. (2019) were reviewed (Table 1). The participants in the intervention groups of the two studies had been doing yoga for a total of 8 weeks, and they were followed up for 4 weeks by Loudon et al. (2014) [24]. However, the frequencies and modes of doing yoga were not the same. The participants had a 90-min weekly teacher-led yoga class and a 45-min daily DVD-led session in Loudon et al. (2014) [24] and three sessions per week, two sessions under the supervision, and one DVD-led session in Pasyar et al. (2019) [25]. Moreover, the mean arm volume assessed by water displacement was not reduced in the intervention group, and there was no statistical difference in upper extremity volume between the two groups after 8 weeks in Pasyar et al. (2019) [25]. Nevertheless, in Loudon et al. (2014) [24], the mean arm volume was significantly reduced in the intervention group, although there was no significant mean change in arm volume, and extracellular fluid between the two groups was detected after 8 weeks. In the follow-up, the mean arm volume of the intervention group increased [24].

### 3.2. Resistance Exercise

Five studies with the intervention of resistance exercise were reviewed (Table 2). In Schmitz et al. (2009), 71 women in the weight-lifting group performed a progressive weightlifting program twice per week in a community fitness center for 12 months, and they were professional-led and group-based for the first 13 weeks [26]. Seventy women in the control group did not change their exercise level during the study. The percentage of women with a change in swelling of 5% or more was similar, and the percentage of mean interlimb volume discrepancy between baseline and 12 months was not statistically different between the two groups [26].

The aim of Kim et al. (2010) was to evaluate the different effects of CDP along with or without active resistance exercise [27]. Twenty patients in the active resistive exercise group did progressive resistance exercise (PRE) for 15 min per day, 5 times per week for 8 weeks after CDP, while twenty patients in the nonactive resistive exercise group completed CDP only [27]. The researchers found that the proximal volume of the affected arm had a notable reduction (*p* < 0.05) in both groups after the interventions [27]. Moreover, the proximal volume of the affected arm had a greater improvement in the active resistive exercise group than in the nonactive resistive exercise group (*p* < 0.05). However, there were no statistical changes in the distal volume and total volume of the affected arm whether between pre- and post-intervention or between the groups after 8 weeks [27].

Jeffs and Wiseman carried out a study in 2013 [28]. Eleven women in the intervention group completed a daily (about 10–15 min) home-based gravity resistive isotonic arm exercise program for 26 weeks, while 13 women in the control group performed self-care only. Limb volume was measured by Perometer. A significant reduction in % excess limb volume (ELV) was found in the intervention group (*p* = 0.013) at week 26 but not in the control group (*p* = 0.1), although there was no great difference between the two groups (*p* = 0.187) [28]. In Bok et al. (2016) [29], 16 patients in the PRE group conducted PRE and CDP, starting from five repetitions of six exercises twice a day and increasing by another five repetitions every week, while 16 patients in the non-PRE group were given CDP only. After 8 weeks, both the proximal and distal circumferences of the affected arm measured by tape had a great decrease (*p* < 0.05) in the PRE group, but this was not found in the non-PRE group [29]. 

Cormie et al. (2013) performed the first RCT to assess the possibility of different intense resistance exercises as management for patients with BCRL [30]. In their study, two experimental groups completed two 60-min sessions per week in a group of 8–10 participants under supervision for 3 months, and one control group performed the usual care [30]. In the experimental groups, 22 women performed high-intensity resistance exercise (10–6 reps of 75–85% of 1 repetition maximums (RM), 1–4 sets per exercise), while 21 women performed low-intensity resistance exercise (20–15 reps of 55–65% of 1RM, 1–4 sets per exercise) [30]. No difference between these groups was found in terms of the swelling of the affected arm, which was assessed by three methods: (1) BIS, (2) dual-energy X-ray absorptiometry, and (3) arm circumference measurements [30].

### 3.3. Complex Exercise

There were 4 RCTs investigating the effect of resistance exercise combined with aerobic exercise on patients with BCRL (Table 3). Nonetheless, the measurements collection and analysis were blinded to the group’s allocation in the 3 of 4 trials, Hayes et al. (2009) [31], Schmitz et al. (2019) [32], and Kilbreath et al. (2020) [33], but this was not mentioned in McKenzie and Kalda (2003) [34]. The exercise plans in these studies were various although all training programs were progressive. However, significant differences in lymphedema severity between the intervention group and the control group were observed in none of the studies.

In McKenzie and Kalda (2003), women completed an 8-week progressive upper extremity complex exercise plan, 3 times per week [34]. Lymphedema severity was measured by water displacement and calculated from arm circumference, but there was a great difference in neither method between the exercise group (n = 7) and the control group (n = 7) [34]. Forty-one women in Kilbreath et al. (2020) completed three 1-h exercise sessions every week (only the first week under supervision), consisting of 10-min warm-up, 30-min resistance training (progressive, moderate to vigorous) and two 10-min vigorous-intensity aerobic sessions, whereas 47 women in the control group were not given any exercise advice [33]. The BIS ratio in the exercise group had a considerable decrease in the breast compared with the control group (*p* = 0.018) but no change in the arms and legs [33].

Hayes et al. (2009) provided a 12-week training program that gradually increased in frequency, intensity, duration, and type, including a total of 20 supervised group-based sessions [31]. The frequency progressed from 3 times to ≥4 times per week, the intensity progressed from low to high, and the duration progressed from 20–30 min per session to ≥45 min per session [31]. Walking, floor-based, and water-based aerobic exercises combined with water-based, free-weight, and machine-weight resistance were carried out. Lymphedema status, which was measured by BIS and Perometry, had no significant difference between the intervention group (n = 16) and the control group (n = 16) during either the 12-week intervention or the 12-week follow-up [31].

Eighty-seven participants in the exercise intervention group in Schmitz et al. (2019) performed two sets of 10 repetitions of 9 resistance exercises, twice per week with dumbbells in the first 6 weeks but progressed to 3 sets thereafter [37]. The weights of dumbbells were tailored to each individual and progressed by 0.45–0.9 kg every 2 weeks from 0.45 kg to 9.45 kg. Moreover, six 90-min weekly classes were offered followed by monthly classes from week 7 [37]. Meanwhile, participants were asked to do aerobic exercise, walking 90 min at week 1–3, and gradually increasing to 180 min per week by week 7 [37]. After this 52-week exercise program, there was no significant difference between the exercise group and the control group in the percentage of interlimb volume differences assessed by Perometry [37].

### 3.4. Water-Based Exercise

Two studies implemented water-based exercise or aqua exercise in the intervention group (Table 4). The participants completed 45-min aqua lymphatic therapy once a week consisting of low-intensity resistance exercise once a week in Tidhar and Katz-Leurer (2010) [35] and a 30-min session of moderate exercise three times a week in Johansson et al. (2013) [36]. Additionally, lymphedema severity was assessed by different methods. The limb volume was measured by using water displacement in Tidhar and Katz-Leurer (2010) [35] but measured by Perometry in Johansson et al. (2013) [36]. In the latter study, BIS was used to measure the impedance of extracellular fluid, and tissue dielectric constant measurement was used to assess local tissue water. However, whichever methods were used in these two studies, neither a vicious incident nor reduced lymphedema severity was reported [35,36]. 

## 4. Discussion

BCRL comes with a multitude of symptoms such as pain, heaviness, tightness, limited range of motion, etc. Localized swelling, which is caused by reserved lymphatic fluid in the interstitial tissue, can be one of the main concerns of cancer survivors with lymphedema [38]. Lymphedema severity may not only affect cancer survivors’ physical fitness but also their mental health, socialization, and career, and eventually, their QoL may also dramatically decline [39]. Unfortunately, lymphedema is incurable so far, but some methods can help manage it [40]. Exercise has been proposed to be safe and effective for managing BCRL in the last twenty years [19]. However, there is limited understanding regarding the effects of various exercise modalities on lymphedema severity.

The effect of exercise on lymphedema severity was shown in 4 of 13 RCTs, three studies with resistance exercise alone [27,28,29], and one study with yoga intervention [24]. In the three studies with resistance exercise alone, a significant reduction of lymphedema severity was observed in the intervention group but not in the control group in Bok et al. (2016) [29] and Jeffs and Wiseman (2013) [28]. In Kim et al. (2010) [27], the reduction was noticed in both groups but was statistically greater in the intervention group than in the control group. On the other hand, the randomization method and measurement blinding were not reported by Kim et al. (2010), and in Bok et al. (2016), the protocol showed that QoL would be assessed by questionnaire, but the results were not mentioned in the following part, leading to a high risk of reporting bias, although this was not the primary concern of this review.

However, not all included studies with resistance exercise alone reported reduced lymphedema severity. In Cormie et al. (2013) [30], no considerable change in the extent of swelling was found. Moreover, although a significant change in the breast was shown between groups in Kilbreath et al. (2020) [33], BIS is an appropriate tool to measure limb swelling but did not work well for assessing breast edema, and there was no great change in the BIS ratio in the limbs. Furthermore, the reduction of lymphedema severity was not reported in any other studies with complex exercise despite it being a combination of resistance exercise and aerobic exercise. There is no evidence showing that aerobic exercise may weaken the effectiveness of resistance training in mitigating lymphedema severity [37]. As such, the reduction of lymphedema severity may not be related to the exercise modality itself, like resistance exercise. When looking at the elements of the exercise programs, we found that the exercises were high-frequency (daily or nearly daily), low-intensity (using 0.5 kg dumbbells or gravity-resistance), and short-duration (10–15 min per session) in all three studies with reduced lymphedema severity.

Similarly, daily yoga sessions can also be used to manage lymphedema symptoms [41], although the relatively high dropout rate had an unclear impact on the result, and Loudon et al. (2014) assumed that the significant reduction of arm volume in the yoga group may be related to its high mean arm value at baseline compared with the control group [24]. On the other hand, there was no reduced lymphedema severity noticed in the other studies without a high-frequency exercise program. Overall, exercise frequency likely plays a significant role in reducing lymphedema severity. Furthermore, this reduction may not be maintained after terminating the training programs. The reduction in lymphedema severity gained in the intervention period disappeared after the four-week follow-up in Loudon et al. (2014) [24]. Therefore, people with BCRL may have to maintain a lifelong dedication to exercising.

Another point that has been discussed is whether exercise programs should be supervised. The previous study suggested that supervision should be applied for safety and lessening fear [42], but this review has different findings. Most exercise programs in the reviewed studies were semi-supervised, and participants completed some supervised sessions at the designated locations and performed the rest of the sessions at home. In Schmitz et al. (2019) [37], there were six weekly professional-led sessions followed by monthly supervised sessions from week 7 to week 52, and no significant difference in lymphedema severity between the intervention and the control groups after 1 year was found. This was partially attributed to inadequate supervision in the study [37]. The researchers claimed that the elements of exercise programs may not get enough attention to be adjusted as needed in the exercise programs with low-frequency supervised sessions [37]. However, reduced lymphedema severity was reported by Jeffs and Wiseman (2013) and Bok et al. (2016) with home-based resistance exercise programs [28,29]. An easy home-based exercise program can increase adherence because participants can self-determine the time and location at which they perform the exercise program [28]. In other words, supervision may not be necessary to reduce lymphedema severity, and patients with BCRL are likely to benefit from a home-based exercise program with some other means, such as a checklist, exercise diary, unexpected calling [29], or instruction sheets provided [28] to increase compliance and adherence. 

However, it should be noted there are some limitations. This is a qualitative review given the limited number of eligible studies, various measuring methods, and high heterogeneity of exercise programs, etc. There are several different definitions of “lymphedema” used for participants’ inclusion criteria in the different studies. Furthermore, the total number of participants in all studies with daily or near-daily exercise is 123, so a larger trial is needed to investigate whether high-frequency exercise can help reduce lymphedema severity.

## 5. Conclusions

By reviewing these 13 RCTs, we noticed that reduced lymphedema severity was shown in four studies with two different exercise types: resistance exercise alone and yoga. After looking at and comparing the elements of exercise programs in these studies, reduced lymphedema severity may be associated with “high-frequency” exercise programs. Additionally, supervision may not be necessary, and home-based exercise programs can be beneficial for cancer survivors to reduce lymphedema severity. Therefore, daily, or nearly daily, exercise at home is recommended. Moreover, reduced lymphedema severity may not be maintained after ceasing the exercise program, so exercise should be a lifelong practice.

## Figures and Tables

**Figure 1 cancers-16-01367-f001:**
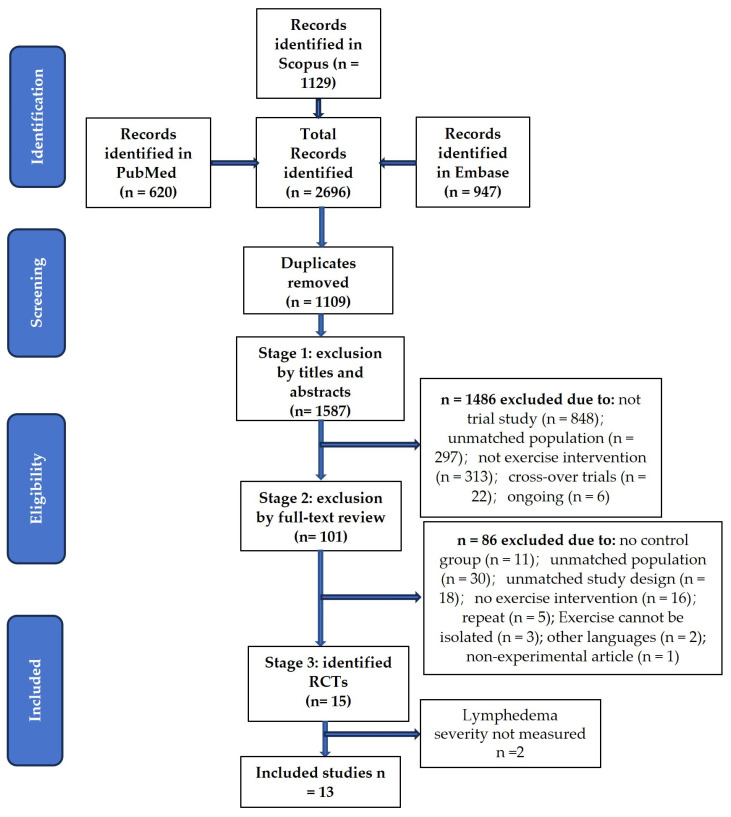
Flowchart of the literature search and the process of selection.

**Figure 2 cancers-16-01367-f002:**
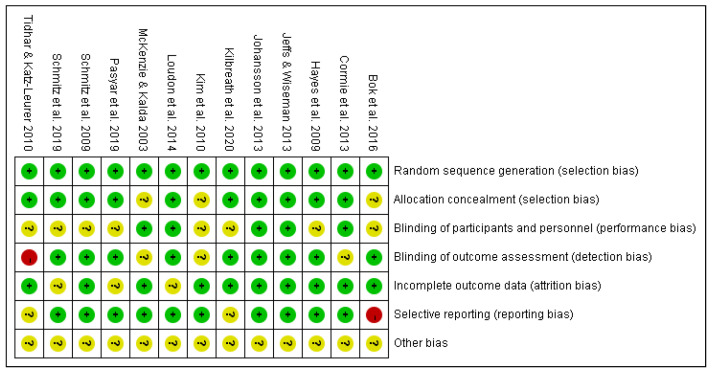
Risk of bias analysis of the included studies [24,25,26,27,28,29,30,31,32,33,34,35,36].

**Table 1 cancers-16-01367-t001:** Participants’ characteristics, intervention details, and outcomes in the studies with Yoga.

Study	No. of Participants and Characteristics	Intervention Details (Frequency, Time, Duration, and Supervision)	Measure and Outcomes
Loudon et al. (2014) [24]	Twenty-eight women (ages 34 to 80) completed breast cancer treatment before ≥6 months, with stage one unilateral BCRL	90-min teacher-led yoga weekly, 45-min DVD-led yoga daily for 8 weeks	No significant mean change in arm volume (assessed by Jobst non-stretch tape), and extracellular fluid (measured by BIS) between the two groups were detected after 8 weeks. The mean arm volume reduced to 72.03 ± 80.77 from 101.45 ± 75.08 in the intervention group after 8 weeks. After another 4 weeks of follow-up, the mean arm volume in the intervention group was increased
Pasyar et al. (2019) [25]	Forty women (>18 years old) with BCRL and an average age of 51.7 years old	Three sessions weekly (two sessions under supervision and 1 session by using DVD at home) for 8 weeks	The mean upper extremity volume was not reduced in the intervention group after 8 weeks. No statistical difference in upper extremity volume was assessed by water displacement between the two groups.

**Table 2 cancers-16-01367-t002:** Participants’ characteristics, intervention details, and outcomes in the studies with resistance exercise.

Study	No. of Participants and Characteristics	Intervention Details (Frequency, Intensity, Time, Duration, and Supervision)	Measure and Outcomes
Schmitz et al. (2009) [26]	One hundred and forty-one patients (average age of 56 ± 9 in the study group and 58 ± 10 in the control group) with stable BCRL after unilateral non-metastatic breast cancer 1 to 15 years	**Frequency and duration:** Twice weekly for 1 year **Intensity, time, progression, and supervision:** in the first 13 weeks, 90-min professional-led sessions in the group, starting from little-to-no resistance and 1–3 new exercises were taught each session (of those weeks, in the first 5 weeks, increasing the number of sets from 2 to 3, after 2 sessions of 3 sets of 10 reps, the resistance were increased gradually), and then unsupervised exercise for the rest 39 weeks	Limb swelling was measured by water displacement. The percentage of women who had a change in interlimb volume difference of 5% or more and the percentage of mean interlimb volume discrepancy between baseline and 12 months were not statistically different between the two groups.
Kim et al. (2010) [27]	Forty women (age from 27 to 76) with unilateral BCRL	**Frequency and duration:** 5 days per week for 8 weeks **Intensity, time, progression:** 6 exercises (seated row, bench press, latissimus dorsi pull-down, 1-arm bent-over row, triceps extension, and biceps curl) with 0.5-kg dumbbells for 15 min; after the first 2 weeks, 1-kg dumbbells were used unless participants felt they were too heavy. **Supervision:** for the first 2 weeks	The volume (measured circumference by tape) of the proximal arm statistically reduced within both groups after 8 weeks. The reduced volume of the proximal arm was significantly greater in the intervention group than in the control group.
Cormie et al. (2013) [30]	Sixty-two women (average age of 56.1 ± 8.1 in the high-load group, 57.0 ± 10.0 in the low-load group, and 58.6 ± 6.7 in the control group) diagnosed with breast cancer at least 1 year before	**Frequency and duration:** twice per week (in a group of 8–10 persons) for 12 weeks **Intensity and supervision:** fully supervised moderate to high-intensity exercise (low-load group: 55–65% of 1 RM using 20–15 RM; high-load group 75–85% of 1 RM using 10–6 RM) **Time:** 1–4 sets per exercise (chest press, seated row/lat pulldown, shoulder press/lateral raise, bicep curl, triceps extension, wrist curl, leg press/leg extension, squat/lunge) for a total of 60 min per session **Progression:** increased in 5–10% increments if participants were able to perform more repetitions than the prescribed RM during a set without worsening arm symptoms.	There was no mean change difference in the extent of swelling (measured by three methods: (1) BIS, (2) dual energy X-ray absorptiometry, and (3) arm circumference measurements) between groups either at the baseline or at 12 weeks.
Jeffs and Wiseman (2013) [28]	Twenty-three women (age from 51 to 73.5) with stable unilateral BCRL ≥ 3 months	**Frequency, time, and duration:** 10–15 min of breathing and gravity-resistive isotonic arm exercises daily for 26 weeks **Supervision:** no	%ELV (calculated from limb volume measured by Perometer) decreased significantly in the intervention group but not in the control group, with no statistically significant difference between the two groups.
Bok et al. (2016) [29]	Thirty-two women (average age of 45.4 ± 8.8 in the PRE group and 53.3 ± 9.54 in the non-PRE group) with unilateralBCRL	**Frequency:** Twice daily for 8 weeks **Intensity and progression:** 5 repetitions of 6 exercises (dumbbell fly, triceps extension, one-arm bent-over row, biceps curl, dumbbell side raise, and lifting thearms forward) with 0.5-kg dumbbells, adding 5 additional repetitions every week **Supervision:** no	Both distal and proximal upper limb circumferences measured by tape reduced significantly in the intervention group after 8 weeks.

**Table 3 cancers-16-01367-t003:** Participants’ characteristics, intervention details, and outcomes in the studies with complex exercise.

Study	No. of Participants and Characteristics	Intervention Details (Frequency, Intensity, Time, Duration, and Supervision)	Measure and Outcomes
McKenzie and Kalda (2003) [34]	Fourteen women (an average age of 56.6 ± 9.0 years old) completed stage I or II breast cancer treatment for at least 6 months with unilateral lymphedema.	**Frequency and duration:** 3 times per week for 8 weeks **Intensity, time, progression, and supervision:** Resistance exercise started from 2 sets of 10 repetitions of 6 exercises (seated row, bench press, latissimus dorsi pull down, one arm bent-over rowing, triceps extension, and bicep curl) with light weights; after the first week, it increased to 3 sets of 10 repetitions, and weights were gradually processed for each participant. The aerobic exercise started from 5–7 min of warm-up aerobic exercise; after 2 weeks, an arm cycle ergometer was used (under supervision), 5 bouts of 1-min cycling at a resistance of 8.3 W and then processed to 20 min of continuous cycling with a resistance of up to 25 W	No statistical change in arm circumference and arm volume (calculated from arm circumference and measured by water displacement) in either group was observed.
Hayes et al. (2009) [31]	Thirty-two women (average age of 59 ± 7 in the intervention group and 60 ± 11 in the control group) with unilateral BCRL at least 6 months	**Frequency and supervision:** 3 times (2 supervised) per week at week 1–4, 4 times (2 supervised) per week at week 5–8, and at least 4 times (1 supervised) per week at week 9–12 **Intensity, time, and progression:** Low-to-moderate-intensity (RPE 3–5) aerobic exercise only for 20–30 min in the first 2 weeks; Low-to-moderate-intensity (RPE 3–5) aerobic exercise and low-intensity (20 reps per exercise) resistance exercise for 20–30 min at week 3 and 4; moderate-intensity aerobic (RPE 4–6) and moderate-intensity (finally completed 15 reps per exercise) resistance exercises for 30–45 min at week 5–8; moderate-to-high-intensity (RPE 4–7) aerobic and moderate- to-high (finally completed 10 reps per exercise) resistance exercises for 45+ minutes at week 9–12	The changes in lymphedema status (measured by BIS and Perometry) were not statistically different between the two groups. No significant change within the groups was observed.
Schmitz et al. (2019) [32]	One hundred and seventy-seven (the number in the control and exercise groups) women (body mass index of 25 to 50) diagnosed with BCRL and completed treatment for at least 6 months, not engaging in resistance training or 3 or more weekly aerobic sessions in the past 1 year, no current weight management surgery or medication taken, no weight loss greater than 4.5 kg in the past 3 months	**Frequency and duration:** Twice weekly for 52 weeks **Intensity, time, and progression:** Two sets of 10 repetitions of 9 resistance exercises with 0.45 to 9.45 kg dumbbells and one 90-min weekly class in the first 6 weeks; from week 7, it increased to 3 sets per session with increased weights by 0.45 to 0.9 kg every two weeks with an upper limit of 9.45 kg and monthly classes provided; aerobic exercise started from 90 min per week, gradually increased walking time to 180 min per week	There was no significant difference between the two groups in the percentage of interlimb volume differences (measured by Perometry) either at the baseline or after 12 months
Kilbreath et al. (2020) [33]	Eighty-nine women (≥18 years old, average age of 59.5 ± 8.0 in the control group and 53.7 ± 10.4 in the exercise group) with BCRL at least 3 months	**Frequency and duration:** Three sessions per week for 12 weeks **Intensity and time:** One-hour exercise session consisted of 10-min warm-up, 30-min resistance exercise, and 20-min aerobic exercise. Resistance exercise: 10–12 repetitions moderate-intensity (5–7 on the OMNI-RES) for weeks 1–6 and high-intensity resistance exercise (7–9 on the OMNI-RES) for weeks 7–12. Aerobic exercise: 60%–85% of HRR, perceived exertion of 11–17. **Progression:** For resistance exercise, the intensity was increased if the weight could be lifted more than 12 times, but if participants could not lift the new weight, the number of repetitions was increased from 12 to 14 and then increased weight. For aerobic exercise, new types of exercise (stationary bike, treadmill, rower, etc.) were added every 4 weeks.	The change in BIS ratio was significant for breasts but not significant for arms and legs between the two group

Note: RPE = Rate of perceived exertion.

**Table 4 cancers-16-01367-t004:** Participants’ characteristics, intervention details, and outcomes in the studies with water-based exercise.

Study	No. of Participants and Characteristics	Intervention Details (Frequency, Intensity, Time, Duration, and Supervision)	Measure and Outcomes
Johansson et al. (2013) [36]	Twenty-nine women (age 56–74 years old) with unilateral BCRL for at least 6 months but without active treatment for the last 3 months	**Frequency, duration, supervision:** three 30-min sessions per week for 8 weeks, supervision only for the first session **Intensity:** moderate intensity (11–13 RPE on the Borg scale)	No change in lymphedema status (measured by Perometry and BIS) was identified.
Tidhar and Katz-Leurer (2010) [35]	Forty-eight women with (average age of 56.2 ±10.7 years old in the study group and 56.5 ± 8.8 years old in the control group) unilateral BCRL and completed intensive CDT for the last 2 weeks	**Frequency, intensity, time, and duration:** one 45-min session of gentle exercises (in a group setting) for 12 weeks **Supervision:** no	No long-term effect of water-based exercise on limb volume (assessed by water displacement) was observed although it reduced limb volume immediately after sessions.

Note: RPE = Rate of perceived exertion.

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
