# Peer review of "The Effect of Exercise on Reducing Lymphedema Severity in Breast Cancer Survivors"

_cancers, 2024, doi:10.3390/cancers16071367_

Round 1

Reviewer 1 Report (Previous Reviewer 2)

Comments and Suggestions for Authors

Dear Authors,

Thank you for work to revise the manuscript and consideration of my previous comments. From my point of view the manuscript now has an optimal structure, content and representation of results. 

Reviewer 2 Report (Previous Reviewer 3)

Comments and Suggestions for Authors

Dear Authors!

While you do not agree with remarks regarding systematic approach that I've made before I appreciate that you have made corrections that made your review nearly systematic. Discussion on strengths and possible weaknesses of included studies makes your conclusions more balanced.

This manuscript is a resubmission of an earlier submission. The following is a list of the peer review reports and author responses from that submission.

Round 1

Reviewer 1 Report

Comments and Suggestions for Authors

"Error! Reference source not found" please remove from text

In introduction a wide background about lymphedema is offered; however, nothing was mentioned  about recent combined surgical and conservative approach to bcrl. The following recent studies could be interesting to be commented PMID: 38086324 and PMID: 36433802.

Paragraph 3.1 unit of measures are missing

Paragraph 3.2. acronyms should be preceeded by the regular form on their first appearance in the text (i.e. PRE and later BIS)

"Hayes et al.’s study" please remove  " 's ". and also later in the text

DIscussion section should be turned to past tense when commenting literature

In discussion section the following systematic reviews should be commented PMID: 35894167  and  PMID: 33840972

Line 296-302 should be moved at the end of discussion section

please check reference 19, author names

    Comments on the Quality of English Language

    The main problem is Discussion section verb tenses. see main report

    Author Response

    Response to reviewer 1:

    Comment 1: "Error! Reference source not found" please remove from text

    Response: Thanks for reminding. It has been moved from the text.

    Comment 2: In introduction a wide background about lymphedema is offered; however, nothing was mentioned about recent combined surgical and conservative approach to BCRL. The following recent studies could be interesting to be commented PMID: 38086324 and PMID: 36433802.

    Response: Thank you for your recommendation. We added the content of combined surgical and conservative approach and referred to these two articles. Please see the 3rd paragraph in the introduction part as tracked.

    Comment 3: Paragraph 3.2. acronyms should be preceded by the regular form on their first appearance in the text (i.e. PRE and later BIS).

    Response: Thanks for reminding. Acronyms are preceded by the regular form on their first appearance in the revision

    Comment 4: "Hayes et al.’s study" please remove " 's ". and also later in the text

    Response: “’s” were all removed.

    Comment 5: Discussion section should be turned to past tense when commenting literature.

    Response: Thank you for pointing this out. We revised the verbs in the discussion to the past tense when commenting on literature.

    Comment 6: In discussion section the following systematic reviews should be commented PMID: 35894167 and PMID: 33840972

    Response: Thank you for your suggestion. These two systematic reviews have been commented on in the discussion part.

    Comment 7: Line 296-302 should be moved at the end of discussion section

    Response: Thank you for your suggestion, Line 296-302 has been moved to the end of discussion section.

    Comment 8: please check reference 19, author names

    Response: Thank you for reminding us. The author’s names were corrected (reference 22 in the revision). 

    Reviewer 2 Report

    Comments and Suggestions for Authors

    Thanks to authors for the narrative review on the effect of exercise on improving lymphedema severity in breast cancer survivors.

    The methodology of the study follow to ground rules for the narrative review. Also, some limitations for the study are identified and mentioned, e.g. various measuring methods to diagnose lymphedema or high heterogeneity of exercise programs seem self-evident based on the broad formulation of the topic. Overall, the study provides a general insight into the application of various exercises to reduce lymphedema related to breast cancer. 

    Few comments:

    1) authors use term "improving lymphedema severity". Perhaps it is appropriate, but based on my experience, term "reducing lymphedema severity" is more widely used (or reducing lymphedema associated symptoms). 

    2) Table 1 (participants characteristics, exercise programs'details and outcomes) is informative but very difficult to follow in too many columns. In some columns information sometimes repeats (e.g. for 1st study supervision is mentioned in frequency description and also in separate column). Perhaps, table could be reviewed and information restructured to reduce the repetitions or even considered representation in two separate tables (e.g. participants and programs and more detailed information on measures/ outcomes in another table). 

    Author Response

    Response to reviewer 2:

    Comment 1: authors use term "improving lymphedema severity". Perhaps it is appropriate, but based on my experience, term "reducing lymphedema severity" is more widely used (or reducing lymphedema associated symptoms).

    Response: Thank you for your comment. We have changed the term “improving lymphedema severity” to “reducing lymphedema severity”

    Comment 2: Table 1 (participants characteristics, exercise programs'details and outcomes) is informative but very difficult to follow in too many columns. In some columns information sometimes repeats (e.g. for 1st study supervision is mentioned in frequency description and also in separate column). Perhaps, table could be reviewed and information restructured to reduce the repetitions or even considered representation in two separate tables (e.g. participants and programs and more detailed information on measures/ outcomes in another table).

    Response: Thank you for your suggestion. We have reviewed the table and restructured the information. The previous table has been split into 4 tables based on four types of exercise in the manuscript.

    Reviewer 3 Report

    Comments and Suggestions for Authors

    This review is positioned as a narrative review that aims to provide advice on what formulation of exercises prescriptions for patients with breast cancer related lymphedema (BCRL) would be beneficial in terms of limb volume reduction. BCRL is an important clinical problem as lymph nodes dissection during cancer surgery puts women at high risk of secondary arm lymphedema development. Most of patients are presented with slight or moderate arm edema in next years after surgery while many suffer from severe edema that causes significant worsening of quality of life and leads to deterioration in physical abilities. While lymphedema treatment is more or less well developed and combines manual lymphatic drainage with strong compression, the possible advantages of physical exercises are still under discussion. Due to that the idea of reviewing data on the impact of exercises on lymphedema severity seems reasonable.

    My main concern about the paper is why authors did not perform a review in a systematic way?

    If the aim was to conduct a narrative review why the data from non-randomized and non-comparative data was not presented and was not discussed? It is now look as a simple short presentation of someone’s RCT data. But there are a lot of data regarding for example different yoga techniques, other exercises not discussed in this review.

    I recommend authors to perform a systematic review with assessing quality of studies, heterogeneity of data, etc.. If meta-analysis is not possible it needs to be explained why.

    Minor remarks

    Lines 84-85. What is meant by “some or all participants at high risk of lymphedema”. If your inclusion criteria was BCRL then all patients had already had lymphedema at inclusion. How the risk could be assessed in them?

    Figure 1. Please, put numbers of excluded records at each stage to the right of the flow.

    Table 1. I recommend to create for table according to four different types of exercises that you’re discussing.

    Lines 296-302. This is not a conclusion. This paragraph has to be moved to the end of a Discussion session.

    Comments on the Quality of English Language

    None

    Author Response

    Response to reviewer 3:

    Comment 1: My main concern about the paper is why authors did not perform a review in a systematic way?

    Response: We appreciate the reviewer’s concern however our aim for this publication was to perform a narrative review summarizing the current findings and knowledge of exercise and its impact on lymphedema in breast cancer patients. This approach will help us to look at studies that varied significantly in study design, type and duration of interventions used and outcome measures.

    Comment 2: If the aim was to conduct a narrative review why the data from non-randomized and non-comparative data was not presented and was not discussed? It is now look as a simple short presentation of someone’s RCT data. But there are a lot of data regarding for example different yoga techniques, other exercises not discussed in this review. I recommend authors to perform a systematic review with assessing quality of studies, heterogeneity of data, etc.. If meta-analysis is not possible it needs to be explained why.

    Response: The purpose of this article is to review and summarize the impact of exercise interventions on lymphedema. We focused on randomized control trials as this would compare between control subjects and intervention participants. As described in Figure 1 findings from 13 RCT trials are discussed. We are aware that there are other types of exercise not reviewed in this paper such as Yoga and Pilates however the impact pf such interventions has not been evaluated in a RCT. The inclusion criteria used for this review article is explained in the methods section of the paper.

    The inclusion criteria were: (i) randomized controlled trials (RCTs), (ii) inclusion of adult participants (>18) living with BCRL, (iii) peer-reviewed and published in the English language, (iv) completed studies with objective measurement of lymphedema severity as one of the outcomes reported, and (v) an organized exercise plan. The exclusion criteria were: (i) some or all participants at high risk of lymphedema, (ii) randomized cross-over control trials, (iii) articles without English translations, (iv) studies with experimental study variables (such as weight control through dietary management) other than exercise, but the effects of exercise cannot be isolated; (v) ongoing experimental studies, (vi) physical therapy as the intervention, (vii) the control group was not given any kind of exercise interventions except for self-care or standard care.

    Minor remarks

    Comment 3: Lines 84-85. What is meant by “some or all participants at high risk of lymphedema”. If your inclusion criteria was BCRL then all patients had already had lymphedema at inclusion. How the risk could be assessed in them?

    Response: We have removed “some or all participants at high risk of lymphedema” from the inclusion criteria.

    Comment 4: Figure 1. Please, put numbers of excluded records at each stage to the right of the flow.

    Response: The numbers of excluded records at each stage has been put to the right of the flow, which is shown in figure 1 in the revision

    Comment 5: Table 1. I recommend to create for table according to four different types of exercises that you’re discussing.

    Response: Thank you for your suggestion. It has been revised and split into 4 tables based on four types of exercise.

    Comment 6: Lines 296-302. This is not a conclusion. This paragraph has to be moved to the end of a Discussion session.

    Response: Thank you for pointing this out. We have moved line 296-302 to the discussion part and m

    Round 2

    Reviewer 1 Report

    Comments and Suggestions for Authors

    The authors responded to the comments of the first round of review

    Author Response

    Thanks you very much we really appreciate your comments. 

    Reviewer 3 Report

    Comments and Suggestions for Authors

    Dear Authors!
    Thank you for addressing some my remarks. The paper improved. But, the main concern was not resolved. I'm insisting on making a systematic review. In a way you presented data they seem to be not fully correct.

    Let me cite your response to my remark: your aim was to perform "narrative review summarizing the current findings and knowledge of exercise and its impact on lymphedema in breast cancer patients. This approach will help us to look at studies that varied significantly in study design, type and duration of interventions used and outcome measures". 
    By excluding other studies than RCTs you then could not achieve your goal as you excluded data from non-randomised and observational studies that might represent real-life practice. Real-life data is important and have to be presented if you claim you were going to summarize current data and knowledge.
    If you use systematic approach you must assess qualities of the studies that you include in the review. This is crucial for drawing correct conclusions. Let's check for example two of RCTs that you analyzed. Loudon et al. reported huge loss for follow-up, i.e., 9 of 28 patients which is more than 30%. Kim et al. did not reported randomization method. QoL between groups was compared without correction for multiple comparisons. Nevertheless, you made conclusions based on the studies with risks of biases and did not reported those risks. Some of this findings must had been discussed as huge loss to follow up in Loudon's study which makes nearly impossible to take into account of reported advantages of yoga.
    Systematic approach would have led to more accurate and cautious conclusions.

    Comments on the Quality of English Language

    -

    Author Response

    Dear Authors!
    Thank you for addressing some my remarks. The paper improved. But, the main concern was not resolved. I'm insisting on making a systematic review. In a way you presented data they seem to be not fully correct.

    Response: We do no aim to complete a systematic review. A systematic review is beyond the scope of this submission. This comment is not specific and therefore cannot be addressed. In addition, other reviewers have not raised any concerns over presentation of data being incorrect. 

    Let me cite your response to my remark: your aim was to perform "narrative review summarising the current findings and knowledge of exercise and its impact on lymphedema in breast cancer patients. This approach will help us to look at studies that varied significantly in study design, type and duration of interventions used and outcome measures". 

    Response: We have decided to include on randomised controlled studied in this review as randomized controlled trials (RCTs) are considered the gold standard in clinical research for evaluating the efficacy and safety of interventions. In a narrative review, which is a type of literature review that provides a summary of a topic through a narrative synthesis of the available evidence, RCTs are preferred for several reasons:

    High Internal Validity: RCTs are designed to minimize bias and confounding factors, thus providing high internal validity. This means that the observed effects can be more confidently attributed to the intervention being studied rather than other factors.

    Controlled Conditions: In RCTs, researchers control the conditions under which the intervention is delivered and the outcomes are measured. This helps ensure that the observed effects are not influenced by external factors.

    Randomisation: Random assignment of participants to treatment and control groups helps to ensure that any differences between the groups are due to chance rather than systematic differences in baseline characteristics. This strengthens the validity of the study's findings.

    Blinding: In many RCTs, blinding (or masking) techniques are used to minimise bias by ensuring that participants, researchers, and outcome assessors are unaware of which participants are receiving the intervention and which are receiving a placebo or alternative treatment. This helps to reduce the potential for biased outcomes assessment.

    Quantitative Analysis: RCTs typically provide quantitative data that can be analysed statistically to assess the magnitude and significance of the intervention's effects. This allows for a more objective evaluation of the evidence compared to qualitative or observational studies.

    Generalisability: While RCTs are often conducted under tightly controlled conditions, they still provide valuable insights into the efficacy and safety of interventions in real-world settings. By including diverse populations and settings, RCTs can help to assess the generalisability of the findings to broader populations.

    By excluding other studies than RCTs you then could not achieve your goal as you excluded data from non-randomised and observational studies that might represent real-life practice. Real-life data is important and have to be presented if you claim you were going to summarize current data and knowledge. 

    Response: In a narrative review, including RCTs allows for a more rigorous and evidence-based synthesis of the literature on a particular topic. By focusing on studies with high methodological quality and strong internal validity, narrative reviews can provide clinicians and researchers with reliable information to inform clinical practice and guide future research efforts. For this reason we have chosen to restrict our study to RCTs. Non-RCTs do not necessarily reveal real life data. We are not evaluating current practice but experimental validation of exercise protocols that are properly monitored and have appropriate controls which can be used to design future exercise prescription for lymphedema that can be further evaluated in feasibility studies. This is in line with our aim below:

    “The aim of this literature review is to extract information from existing studies to provide some advice for the design of exercise prescriptions in terms of reducing lymphedema severity.”

    We have not claimed to provide a full and complete summary of current data and knowledge.  

    If you use systematic approach you must assess qualities of the studies that you include in the review. This is crucial for drawing correct conclusions. Let's check for example two of RCTs that you analysed. Loudon et al. reported huge loss for follow-up, i.e., 9 of 28 patients which is more than 30%.

    Kim et al. did not reported randomization method. QoL between groups was compared without correction for multiple comparisons. Nevertheless, you made conclusions based on the studies with risks of biases and did not reported those risks. Some of this findings must had been discussed as huge loss to follow up in Loudon's study which makes nearly impossible to take into account of reported advantages of yoga.

    Response: We have addressed these issues in the discussion/results section as follows:

    (All changes in V3.0 of the manuscript text submitted is highlighted in green) 

    Systematic approach would have led to more accurate and cautious conclusions.

    Response: We do not agree with the reviewer that a systematic approach is needed or superior. A systematic review is beyond the aim and scope of this submission.